# Effects of 120 vs. 60 and 90 g/h Carbohydrate Intake during a Trail Marathon on Neuromuscular Function and High Intensity Run Capacity Recovery

**DOI:** 10.3390/nu12072094

**Published:** 2020-07-15

**Authors:** Aritz Urdampilleta, Soledad Arribalzaga, Aitor Viribay, Arkaitz Castañeda-Babarro, Jesús Seco-Calvo, Juan Mielgo-Ayuso

**Affiliations:** 1Centro Investigación y Formación ElikaSport, Cerdanyola del Valles, 08290 Barcelona, Spain; a.urdampilleta@drurdampilleta.com; 2Institute of Biomedicine (IBIOMED), Physiotherapy Department, University of Leon, Campus de Vegazana, 24071 Leon, Spain; marisolarribal@gmail.com; 3Glut4Science, Physiology, Nutrition and Sport, 01004 Vitoria-Gasteiz, Spain; aitor@glut4science.com; 4Health, Physical Activity and Sports Science Laboratory, Department of Physical Activity and Sports, Faculty of Psychology and Education, University of Deusto, 48007 Bizkaia, Spain; arkaitz.castaneda@deusto.es; 5Institute of Biomedicine (IBIOMED), Physiotherapy Department, University of Leon, Researcher at the Basque Country University, Campus de Vegazana, 24071 Leon, Spain; dr.seco.jesus@gmail.com; 6Department of Biochemistry, Molecular Biology and Physiology, Faculty of Health Sciences, University of Valladolid, 42004 Soria, Spain

**Keywords:** resistance, carbohydrates, fatigue, recovery, gut training, performance, gastrointestinal discomfort, absorption

## Abstract

Background: Current carbohydrate (CHO) intake recommendations for ultra-trail activities lasting more than 2.5 h is 90 g/h. However, the benefits of ingesting 120 g/h during a mountain marathon in terms of post-exercise muscle damage have been recently demonstrated. Therefore, the aim of this study was to analyze and compare the effects of 120 g/h CHO intake with the recommendations (90 g/h) and the usual intake for ultra-endurance athletes (60 g/h) during a mountain marathon on internal exercise load, and post-exercise neuromuscular function and recovery of high intensity run capacity. Methods: Twenty-six elite trail-runners were randomly distributed into three groups: LOW (60 g/h), MED (90 g/h) and HIGH (120 g/h), according to CHO intake during a 4000-m cumulative slope mountain marathon. Runners were measured using the Abalakov Jump test, a maximum a half-squat test and an aerobic power-capacity test at baseline (T1) and 24 h after completing the race (T2). Results: Changes in Abalakov jump time (ABK_JT_), Abalakov jump height (ABK_H_), half-squat test 1 repetition maximum (HST_1RM_) between T1 and T2 showed significant differences by Wilcoxon signed rank test only in LOW and MED (*p* < 0.05), but not in the HIGH group (*p* > 0.05). Internal load was significantly lower in the HIGH group (*p* = 0.017) regarding LOW and MED by Mann Whitney *u* test. A significantly lower change during the study in ABK_JT_ (*p* = 0.038), ABK_H_ (*p* = 0.038) HST_1RM_ (*p* = 0.041) and in terms of fatigue (*p* = 0.018) and lactate (*p* = 0.012) within the aerobic power-capacity test was presented in HIGH relative to LOW and MED. Conclusions: 120 g/h CHO intake during a mountain marathon might limit neuromuscular fatigue and improve recovery of high intensity run capacity 24 h after a physiologically challenging event when compared to 90 g/h and 60 g/h.

## 1. Introduction

Participation in ultra-endurance mountain race events (>4 h) has increased in recent years [1]. The different distances run by participants range from mountain marathons (42,195 m) to multistage ultra-marathons (up to 350 km), with an accumulative altitude gain of 24,000 m during the most extreme events [2]. In addition, mountain athletes are exposed to different environmental conditions, such as irregular terrains with a variety of geographical and topographic characteristics, climatic conditions, altitude exposure and temperature fluctuations [2,3,4]. This results in extreme physiological demands which may cause, among other things, negative energy balance, dehydration, decrease in blood glucose levels, muscle and hepatic glycogen depletion, exercise induced muscle damage (EIMD) and inflammation [2,3,4,5], and therefore might induce high levels of neuromuscular fatigue [2,6]. In this sense, the fatigue experienced by the runner could be quantified by monitoring the internal exercise load (determined by intensity (measured or perceived) × time) endured by an athlete during exercise [7,8] and by looking for different strategies that allow it to be delayed [7].

Carbohydrate (CHO) intake during endurance exercise has been shown to delay neuromuscular fatigue and improve exercise capacity and work rate significantly in a dose–response relationship [9,10]. In this sense, although the current recommendations in events lasting more than 2.5 h include the 90 g /h CHO intake [11,12], a gut training avoids gastrointestinal discomfort that it could facilitate the intake of greater amounts of CHO to the recommendations [13]. In this sense, although >90 g/h CHO intake may have controversial results [9,14], Pfeiffer et al. showed that athletes who consumed 120 g/h were among the fastest during two ultraendurance events, indicating a delay in the onset of fatigue [10]. In addition a study carried out by our lab has recently demonstrated that higher CHO intake (120 g/h) than recommended could be a determining factor in the internal exercise load response and could limit exercise-induced muscle damage (EIMD) in elite trail runners 24 h after completing a mountain marathon, suggesting that recovery time after such an endurance event could be shortened by a suitable CHO intake during exercise [15]. In addition, it is well known that CHO substrate availability plays a central role in peripheral fatigue, but also within the central nervous system and, thus, in central fatigue [16]. Stewart et al. [17] showed that glucose intake (1.23 ± 0.11 g/kg body mass) during exercise with a 15-min frequency in 15 untrained participants improved muscle function due to attenuated disturbances in the membrane excitability, suggesting that peripheral fatigue could be delayed by CHO intake during exercise.

Peripheral fatigue is understood as the failure of local mechanisms in the muscle and, therefore, the decrease in the contraction and relaxation function that are related with the energetic status of the muscle cell [18,19]. Among these mechanisms, the potential action in the sarcolemma, the excitation–contraction (E-C) coupling and the interaction between actin and myosin proteins that allowed the muscle to contract and relax [20]. In this sense, the link between localized intramyofibrillar glycogen content and muscle function, mediated by the Ca^2+^ release from the sarcoplasmic reticulum (SR), has been established in the literature [21,22]. A study conducted on elite cross-country skiers showed a correlation between the reduction in skeletal muscle glycogen and release rate of Ca^2+^ after completing 1 h of maximum effort [22]. Moreover, significant differences were found in glycogen content and Ca^2+^ release after 4 h post-exercise between the group that consumed CHO (1 g/kg body weight/h) and the placebo one (water), suggesting that CHO intake during exercise might take on a major role in improving short term glycogen replenishment and muscle function [22].

Additionally, both short- and long-term recovery periods following exercise play an important role in ensuring a suitable return to physiological and metabolic homeostasis in athletes. When the neuromuscular function has been affected, replenishment of glycogen stores is required to restore muscle function [23,24]. After exhaustive exercise and glycogen depletion, 36–48 h of high CHO diet (>7–9 g/kg body mass-BM) is needed to compensate muscle glycogen content [25]. However, when EIMD takes place, replenishment capacity is highly compromised, delaying this process by up to 10 days [26]. Moreover, the recovery kinetics of the neuromuscular function were studied following an ultra-marathon race, showing interesting results regarding peripheral and central fatigue [3]. While maximal voluntary activation and low frequency fatigue were recovered within 2 days, plasmatic Ca^2+^ was still reduced at this point and muscle damage biomarkers returned to the baseline on day 5 [3]. In general, authors found that the majority of central and peripheral function indexes were recovered within 9 days [3], although it is generally accepted that central fatigue persists longer than peripheral fatigue [27]. Furthermore and with regard to glycogen replenishment and Ca^2+^ release from the SR, Ortenblad et al. [22] found that after ingesting a high CHO diet within 22 h post-exercise, both glycogen content and the Ca^2+^ release rate returned to baseline levels.

Optimal nutritional protocols to ensure recovery following exercise have been extensively documented in the literature regarding glycogen resynthesis, protein synthesis and rehydration [25,28,29,30]. Although few studies have been conducted with the ingestion of protein and carbohydrates during the exercise with the aim of improving recovery [31,32,33], ingested CHO quantities were lower than currently recommended [11]. To the best of the authors’ knowledge, no research has been conducted using only CHO in high doses during exercise to optimize post-exercise recovery, which could prove very interesting when it comes to improving training capacity and performance in multi-stage competitions. As CHO intake represents a possible methodology not only to help improve performance during exercise, but also to improve recovery through different mechanisms such as limiting EIMD, decreasing internal exercise load and neuromuscular fatigue, maintaining suitable levels of blood glucose and sparing muscle and hepatic glycogen, the purpose of this research was to analyze the effects of a high CHO (120 g/h) intake during a mountain marathon on 24 h recovery in elite runners in terms of neuromuscular function and high intensity run capacity in elite male ultra-endurance athletes.

The main hypothesis of this research was to ascertain whether 120 g /h CHO intake could reduce neuromuscular fatigue by internal exercise load and improve long term recovery compared to current recommendations for ultra-endurance events (90 g/h) [34,35], and regular CHO intake on the part of athletes during such races (60 g/h) [36].

## 2. Materials and Methods

### 2.1. Participants and Experimental Protocol 

The current research was put together as a randomized trial with the aim of examining the effects of 120 g/h of CHO intake during a trail marathon race on internal exercise load during exercise and muscle recovery 24 h following exercise. The 120 g/h of CHO supplementation was compared to international references for ultra-endurance events (90 g/h) [34,35], and regular CHO intake of athletes during such races (60 g/h) [36]. The use of 120 g/h was determined because previous studies have shown that after a training of gut tract it is possible to tolerate this amount [10,37] and because 120 g/h of CHO during an ultraendurance race has shown a lower internal exercise load and lower EIMD [15].

Thirty-one elite male athletes were assessed for eligibility in this study (including 2 trail skyrunner world champions from the International Trail Running Association (ITRA) and the International Association of Athletics Federations (IAAF)). After removing 5 athletes because they failed to meet the inclusion criteria (>5 years’ experience in ultra-endurance training, having undertaken personalized training of the gut tract to enhance CHO absorption capacity and tolerance and not taken any drugs or performance supplements [38] to avoid any possible interference in the recovery process during the 1-week period prior to the race), 26 athletes were involved in the randomization process and went on to take part in the mountain marathon race. These 26 athletes were inspected by a medical doctor prior to the study in order to check they had no injury or disease. None of the runners were suffering from any disease, and none of them were taking any medication. 

The runners were randomized into three different groups by an independent statistician via a randomization sequence using SPSS software as follows: (I) group that was supplemented with 60 g/h of CHO (LOW; *n* = 8; BMI: 23.0 ± 2.9 kg/m^2^), (II) group that was supplemented with 90 g/h of CHO (MED; *n* = 9; BMI: 22.4 ± 2.6 kg/m^2^) and (III) group that was supplemented with 120 g/h of CHO (HIGH; *n* = 9; BMI: 22.1 ± 3.0 kg/m^2^). The runners were instructed that as soon as they felt any damage and/or gastrointestinal distress which might compromise their athletic performance, they should withdraw from the marathon to avoid interference in the results. As a result, 6 athletes withdrew during the race (3 with gastrointestinal discomfort–flatulence and/or reflux and 3 with injury). The remaining athletes finished the marathon without experiencing any injuries and/or gastrointestinal difficulties. Consequently, the sample included in the current research comprised 20 athletes, including 2 world champions (6 runners for the LOW, 7 runners for the MED and 7 runners for the HIGH) [15].

All participants ingested CHO during the race using the same gels made for this research at the University of Valladolid Physiology Laboratory (Soria) by an experienced pharmacist using a gel packaging machine. The gels contained 30 g of maltodextrin (glucose) and fructose (ratio 2:1) to increase exogenous CHO oxidation during exercise [39]. The athletes carried previously configured GPS alarms to notify them that they should intake CHO gels every 15, 20 and 30 min during the mountain marathon race, with runners needing to consume 1/4, 1/3 or 1/2 of the total g of CHO per hour according to their group (HIGH, MED, LOW, respectively) (Figure 1). Although athletes drank water *ad libitum* during the race, the participants ingested no food other than the gels. The athletes were instructed to extract the maximum content of the gels so that the residual amount was minimal. Likewise, all the athletes who finished the race confirmed through a questionnaire that all the gels had been taken at the time and in the manner previously indicated.

The “marathon of Oiartzun” is a mountain marathon race (42.195 km) which began at 9:00 am in Oiartzun (Guipúzcoa-Spain) (10 °C, 60% humidity and 10 km/h wind speed) and was controlled by official chronometers. The race consisted of an entry and exit to a circuit that the runners had to complete 3 times. The height accumulated gradient of the mountain marathon race was 3980.80 m (1990.40 m positive and negative, respectively) (Figure 1), while the maximum and minimum height was 638.20 m and 3.80 m, respectively.

During the race, the heat rate (HR) was documented using a HR monitor, and the mean HR (HR_mean_) and maximum HR (HR_max_) during the event were also recorded.

Prior to commencement of the study, all athletes received information about the purpose of it and associated risks and benefits. Subsequently, everyone signed an informed consent. The study was designed according to the guidelines laid out in the Declaration of Helsinki and approved by the Human Ethics Committee at the Valladolid Health Area, Valladolid, Spain (PI 19-1345).

### 2.2. Dietary Assessment

By way of an inclusion criterion, it was stated that all athletes should have undertaken gut training which involved ensuring >90 g/h CHO intakes > 2 days/week over the 4 weeks prior to the race [13,40].

Regarding diets and menus during the research period, all were put together individually for each athlete in accordance with international recommendations for ultra-endurance sports [34,35] by the same certified sport dietitian-nutritionist.

The athletes obtained a personalized diet for a 48-h period prior to T1 and marathon race day with 9 g CHO/kg BM/day, 1.5 g protein/kg BM/day and 0.5 g fat/kg BM/day to adjust their glycogen content. This diet comprised, among other foods, vegetables, olive oil and fish, although it did not contain any fatty meat or butter [41]. Moreover, runners had breakfast (2 g CHO/kg BM) at the ElikaEsport Health Center 3 h prior to commencement of the mountain marathon race. This breakfast consisted of rice or paste, corn cereals with oat drink, cooked fruit and biscuits with jam, sweet quince or cheese.

At the end of the marathon, each athlete ingested 1.2 g of CHO/kg BM with 0.3 g of whey isolate protein/kg BM in the recovery shakes.

Within the next 24 h until T2, each runner consumed a suitable diet (9g CHO/kg BM, 1.5 g protein/kg BM and 0.5 g fat/kg BM) in order to replenish glycogen levels to almost 90–93% of previous muscle glycogen [25,42].

### 2.3. Athletic Performance Test

Runners attended the laboratory for an athletic performance test the week prior to the trail marathon race (T1) and 24 h after it (T2). The internal exercise load was estimated by training impulse (TRIMP) and athletic recovery was assessed using the Abalakov jump test (ABK), Drop jump test (DJ), Half Squat test (HST) and aerobic power-capacity test.

The two test sessions were carried out at the ElikaEsport Health Center under standard conditions (temperature: 20 °C and humidity: 55%) for both test sessions, and the tests were completed following a standardized 15-min warm-up. The warm-up involved 10 min of running with two 1-min speeding up (at 3 min and 5 min) and 5 min of injury prevention drills consisting of general movements, dynamic/static stretching and core stability.

#### 2.3.1. Neuromuscular Function

Abalakov jump test [43] and half squat [44] test were chosen to measure the neuromuscular function of leg extensor muscles in athletes and recreationally active men given that they can achieve this with a high degree of reliability.

Abalakov jump test: Participants made 3 countermovement with 30-s breaks between jumps [45]. All runners had to start from an upright position and performed a knee flexion of 90° followed by an extension as fast as possible to reach the highest possible jump height. An optical (infrared) data collection was used (Optojump Next Microgate, Bolzano, Italy) to measure jump time (ABK_JT_), and Abalakov jump height (ABK_H_) was calculated [46]. The best of the three records was used for statistical analysis.

Half Squat test: 5-min following Abalakov jump test, runners performed a 1-repetition maximum test (1RM) of half squat test (HST_1-RM_) using a Multipower machine (BH Max Rack LD400, Vitoria, Spain). After 5 min rest from HST_1-RM_ the runners performed three repetitions at maximal speed for a load of 70% of their 1-RM in this half-squad exercise, with 1-min rest between repetitions to determinate the concentric movement speed of half-squad test (HST_Speed_). During half squat test the athletes performed a knee flexion until the thigh was parallel to the ground (90° knee angle) and, following a command, moved the bar up as fast as possible, without their shoulder losing contact with the bar [47]. They then had to wait 3 s to remove the elastic component, after which they were given an external signal to perform a concentric extension of the lower limbs until reaching 180° at the highest possible speed. The average concentric movement speed was measured using the PowerLiftversion 4.2.4 app for Iphone [48].

#### 2.3.2. High Intensity Run Capacity

Aerobic power-capacity test: Runners performed a maximum aerobic power test on an ergometric tape (Tunturi Pure 6.1, Almere, The Netherlands). The protocol used involved maintaining a constant speed of 20 km/h, with a slope of 1%—this being the maximum possible time until exhaustion [46]. During the test, the maximum HR (HRmax) was recorded using HR monitors (Polar V800, Kempele, Finland) taking as reference the highest HR achieved by the athletes during the test. Likewise, at the end of this test a blood sample was obtained from the earlobe to detect blood lactate using a Lactate Scout analyzer (SensLab GmbH, Leipzig, Germany), and at the end of the test athletes also indicated the rate of perceived exertion (RPE) based on the Borg scale [49].

### 2.4. Internal Exercise Load

During the mountain marathon racing, the heat rate (HR) of all runners was monitored continuously using the same HR monitors as used in the aerobic power-capacity test. Likewise, during the race the HR_mean_ and HRmax were recorded, and internal exercise load was also calculated using individualized training impulse (TRIMP) [49,50]. TRIMP was obtained as the product of trail marathon duration and intensity multiplied by a nonlinear metabolic adjustment factor. For its part, the trail marathon intensity was calculated as: ΔHR = (HR_mean_ − resting HR)/(HR_max_ − resting HR).

### 2.5. Anthropometry and Body Composition

All anthropometric and measurements were administered by an internationally certified level 3 anthropometrist. The anthropometrist administered measurements at T1 in accordance with the International Society for the Advancement of Kineanthrometry (ISAK). All measurements were taken in duplicate, with the exception of those that exceeded 5% difference between each other, which required a third measurement.

BM (kg) and height (cm) were measured using a scale (Mod. 220; SECA Medical, Bradford, MA, USA), with 0.1 kg and 1 mm precision respectively. Body Mass Index (BMI) was considered in accordance with the equation: BM/height^2^ (kg/m^2^), while triceps, subscapular, suprailiac, abdominal, front thigh and medial calf skinfolds were examined using a skinfold caliber (Harpenden Skinfold Caliber, British Indicators Ltd., London, UK), with 0.2 mm precision. The sum of these 6 skinfolds (mm) was then calculated. A Lufkin model W606PM measuring tape with 1 mm precision was used to measure the perimeters (relaxed arm, mid-thigh and calf) in cm. These girths were all corrected for the skinfold at the site using the following formula: (corrected girth = girth − (π × skinfold thickness at the site)). Relaxed arm girth was corrected for triceps skinfold (CAG), mid-thigh girth corrected for front thigh skinfold (CTG), and calf girth was corrected for medial calf skinfold (CCG). Muscle mass (MM) was predicted using the Lee [51] equation for males and Caucasian athletes:MM = height × (0.00744 × CAG^2^ + 0.00088 × CTG^2^ + 0.00441 × CCG^2^) + 2.4 − 0.048 × age + 7.8.

### 2.6. Statistical Data Analyses

Statistical analysis was completed by Statistical Package for the Social Sciences 24.0 (SPSS Inc., Chicago, IL, USA), with results being shown as mean and standard deviation. The significance level for all analyses was set at *p* < 0.05.

Although the data obtained presented a parametric distribution after using the Shapiro–Wilk test (*n* < 50), non-parametric tests were performed because the sample in each of the study group was very small. The percentage changes of the physical test between T1 and T2 were considered as Δ (%): ((T2 − T1)/T1) × 100. Δ (%) of athletic recovery (Abalakov jump test, half-squad test and aerobic power-capacity test) was compared among 3 CHO intake groups using Kruskal–Wallis test with the CHO intake groups as the fixed factor. A Mann Whitney *u* tests test was completed for pairwise comparisons between groups.

Similarly, differences between T1 and T2 in each athletic recovery test in each CHO intake group were assessed using a Wilcoxon signed rank test.

## 3. Results

Table 1 shows race time, age, body composition and anthropometric characteristics of subjects at T1 according to each study group. No significant differences were observed for race time, age or anthropometric characteristics between groups (*p* > 0.05).

Table 2 shows the values obtained from the Abalakov and half-squat test at T1 and T2 in the three study groups. A significant decrease was noted in ABK_JT_ and ABK_H_ in LOW and MED (*p* < 0.05), while the HIGH did not show any significant differences in these parameters between T1 and T2 (*p* > 0.05).

On the other hand, although there was a tendency towards a smaller loss in HST_1-RM_ in HST_Speed_ in the HIGH (HST_1-RM_: T1: 97.17 ± 8.60 vs. T2: 90.09 ± 14.20 Kg; HST_Speed_: T1: 0.57 ± 0.14 vs. T2: 0.55 ± 0.18 m/s) in terms of LOW (HST_1-RM_: T1: 103.32 ± 35.67 vs. T2: 81.66 ± 32.42 Kg; HST_Speed_: T1: 0.65 ± 0.06 vs. T2: 0.60 ± 0.11 m/s) and MED (HST_1-RM_: T1: 109.07 ± 33.62 vs. T2: 91.91 ± 25.54 Kg; HST_Speed_: T1: 0.63 ± 0.10 vs. T2: 0.50 ± 0.09 m/s), no significant differences were observed in the group-by-time for these parameters (*p* > 0.05). However, there were significant declines in HST_Speed_ in MED between T1 and T2 (*p* < 0.05).

Table 3. Sets out the aerobic power-capacity test results of the three groups at T1 and T2. A significant decrease was shown in aerobic power-capacity test time in MED between T1 (104.0 ± 48.1 s) and T2 (87.9 ± 39.4 s) (*p* < 0.05). Moreover, although there was a tendency for a decreased lactate in MED (T1: 5.80 ± 0.91 vs. T2: 4.79 ± 1.35 mmol/l; *p* < 0.05)), only LOW evidenced a significant reduction in lactate between T1 (7.45 ± 1.42 mmol/l) and T2 (5.65 ± 1.27 mmol/l) (*p* < 0.05). However, HR max showed significant declines both in LOW and MED between T1 and T2 (*p* < 0.05). On the other hand, Borg showed a significant decrease in HIGH throughout the study (T1: 18.29 ± 0.76 vs. T2: 17.00 ± 1.00; *p* < 0.05), and the Borg value in T2 was significantly less compared to LOW (18.83 ± 0.98) and MED (18.71 ± 1.38) (*p* < 0.05).

Figure 2 shows the TRIMP during the trail marathon, with a significant difference in TRIMP being evidenced between groups (*p* = 0.017). Specifically, a significant higher TRIMP was observed in LOW (399.8 ± 17.5) and MED (371.2 ± 16.2) compared to HIGH (314.8 ± 16.2).

Figure 3 displays the percentage of change of ABK and HST between T1 and T2 in the three study groups. Significant differences were observed among groups in ABK_JT_ (*p* = 0.038) and ABK_H_ (*p* = 0.038). In this sense, there were significant improvements (*p* < 0.05) in ABK_JT_ in the HIGH (0.53 ± 5.12%) compared to LOW (−5.90 ± 4.18%) and MED (−5.02 ± 4.22%). Likewise, there was a significant improvement (*p* < 0.05) in ABK_H_ in the HIGH (1.19 ± 8.05%) regarding LOW (−11.33 ± 8.00%) and MED (−9.66 ± 8.19%). Figure 4 also shows significant percentage differences in HST_1-RM_ among groups between T1 and T2 (*p* = 0.041). Specifically, a significant smaller decline was noted (*p* < 0.05) in HST_1-RM_ in HIGH (−2.35 ± 7.23%) compared to LOW (15.30 ± 7.54%) and MED (−13.84 ± 9.74%) between T1 and T2, while HIGH (−4.19 ± 11.31%) evidenced a significant smaller decline (*p* < 0.05) in HST_Speed_ compared to MED (−20.06 ± 14.75%) during the study.

Figure 4 shows significant differences in percentage change of time in aerobic power-capacity test (*p* = 0.018) between T1 and T2. Specifically, HIGH (1.28 ± 8.55%) showed a significantly better change in time in aerobic power-capacity test compared to LOW (−14.09 ± 14.98%) and MED (−14.87 ± 11.86%) between T1 and T2 (*p* < 0.05). Likewise, HIGH (1.69 ± 11.78%) displayed a significantly higher increase in lactate (*p* = 0.012) compared to LOW (−22.69 ± 9.38%) and MED (−13.94 ± 12.41%). Furthermore, although the percentage change in HIGH showed a tendency towards better values in Borg compared to LOW and MED, these differences were not significant (*p* = 0.066).

## 4. Discussion

The main purpose of this study was to analyze the effects of 120 g/h CHO intake during a mountain marathon on long-term recovery and compare it to current international recommendations for endurance exercise (90 g/h) [34,35] and regular CHO intake by athletes during ultra-endurance events (60 g/h) [36]. It was hypothesized that a greater amount of CHO ingestion during exercise could result in better recovery parameters, allowing runners to optimize their performance in the 24 h following a race. To this end, this study focused on measuring two main physiological and metabolic qualities involved in the trail running discipline that are determinants for performance: neuromuscular function and high intensity run capacity [3,46,52]. The main findings showed that neuromuscular function, measured by an Abalakov jump test, had a better response 24 h following the mountain marathon race in the HIGH (120 g/h) group compared to MED (90 g/h) and LOW (60 g/h). Moreover, HIGH significantly limited HST performance loss in terms of LOW and MED. Regarding high intensity run capacity, runners that consumed 120 g CHO/h during the mountain marathon evidenced significantly higher lactate production and better performance in the aerobic power-capacity test carried out 24 h after the race compared to MED (90 g/h) and LOW (60 g/h). These results suggest that a higher CHO intake than recommended during a mountain marathon might improve long-term recovery of neuromuscular function and high intensity run capacity and limit the decrease in performance capacity 24 h following a challenging race.

Mountain running events such as marathons and ultra-marathons represent extreme physiological and metabolic challenges as they involve long distances and high exercise intensities, combined with a wide variety of up and down terrain [2]. The runners may find their performance is limited during these events, among other reasons owing to negative energy balance, dehydration, decrease in blood glucose levels, muscle and hepatic glycogen depletion, exercise induced muscle damage and inflammation, and neuromuscular fatigue [2,3,4,5]. In order to face these major challenges to delay fatigue and improve performance, runners should be properly prepared in both neuromuscular and metabolic capacities [3,6] and use several nutritional protocols [12]. Some authors have shown that CHO intake during high load exercise, such as occurs during a mountain marathon, has beneficial effects on performance, fosters faster muscle function and delays fatigue [17,34,53,54,55]. In the present study, although performance did not show any significant improvement, it did show a tendency for runners who ingested 120 g/h of CHO during the marathon race to be faster than those who took 60 g/h and 90 g/h. In addition, runners who took more CHO (HIGH) during the marathon evidenced a lower internal exercise load (TRIMP) compared to the other groups (LOW and MED), which might indicate that the HIGH group suffered less fatigue [7,8].

Although fatigue is multifactorial, neuromuscular fatigue represents a determining factor in mountain endurance events [3,20], and both glycogen content and high intensity run capacity are also supposed to be determinants in endurance and high intensity exercises [12,34,42]. In this study, a significantly lower TRIMP was reported in those athletes that consumed 120 g CHO/h compared with 90 and 60 g/h. Moreover, HIGH showed better neuromuscular function and high intensity running capacity after 24 h relative to LOW and MED, suggesting that the intake of CHO could impact positively on these performance parameters. Neuromuscular fatigue comes when the force production in the muscle is lowered due to disturbances in the central and peripheral nervous system function [19]. Specifically, peripheral fatigue is related to the local muscle mechanisms that impair muscle contraction and relaxation [18,19]. Briefly, distal or peripheral fatigue involves three main components of the muscle function as follows: (I) the transmission of potential action from the sarcolemma, (II) the excitation–contraction (E-C) coupling regulated by the Calcium (Ca^2+^ release from the sarcoplasmic reticulum (SR)) and (III) the interaction between actin-myosin proteins [20]. In this respect, EIMD is an important factor that limits the muscle function affecting these three main components [27,56,57]. Although none of these where directly measured in present study, the results showed previously by our lab where lower EIMD was found with 120 g CHO/h intake [15], could explain the significantly better neuromuscular function recovery reported in this study HIGH group.

Moreover, it has been shown that mountain running endurance events have a high impact on peripheral fatigue, reducing muscle function and performance [3,20,58]. Specifically, changes in excitation–contraction (E-C) coupling failure and neuromuscular propagation reduction may contribute to a decrease in maximal force production [58]. In the same line, downhill running induced eccentric contractions and EIMD are related to so-called low-frequency fatigue (LFF), which has been closely associated with E-C coupling failure and, as explained, with muscle force reduction [20,59,60,61]. In fact, E-C coupling failure results in a decrease in free Ca^2+^ concentration in the cytosol mediated by the reduced Ca^2+^ release from the SR [61]. This Ca^2+^ reduction could be explained by three main factors: (I) failure in translation of the neural command in the plasma membrane [62], (II) uncoupling between the calcium release channels and depolarization signaling in the T-tubule and (III) the depletion of the intramyofibrillar glycogen [22]. The results obtained in this study could be explained by the delay on the glycogen depletion induced by a higher CHO intake during exercise, as previously reported in the literature [22,63]. In the current study, it has been demonstrated that 120 g/h of CHO evidenced a significantly better neuromuscular function 24 h post-exercise compared to 90 g/h and 60 g/h intakes. Moreover, although peripheral neuromuscular function was not directly measured, runners’ high intensity run capacity and lactate concentrations were significantly better in the HIGH group, suggesting that protection of membrane excitability could be a possible reason behind understanding these results. As previously reported, CHO supplementation during exercise has evidenced improvements in muscle function with attenuation of the muscle membrane excitability disturbances [17], suggesting that CHO intake might protect the peripheral neuromuscular function. Along these lines, it is known that higher concentrations of lactate in the muscle improve membrane excitability and, therefore, muscle function [64]. As lactate is the main product of glycolysis, it is mainly stimulated by a higher glycolytic flux which, at the same time, depends on glycogen content and glucose availability [65]. Thus, it is reasonable to hypothesize that a higher exogenous CHO intake, together with the better high intensity run capacity maintained as a consequence, could delay neuromuscular fatigue. Moreover, CHO intake during exercise has been shown to improve short-term muscle function recovery due to a greater intramyofibrillar glycogen content and sarcoplasmic Ca^2+^ release [22].

As the athletes in the HIGH group had significantly higher blood lactate concentrations and it was correlated, at the same point, with a significantly greater high intensity running capacity 24 h post-marathon, it is possible to state that they showed a higher glycolytic capacity. In endurance events, the metabolic flexibility, which is described as referring to the capacity to use different substrates according to acute metabolic demands of the exercise, constitutes a determining factor in exercise performance and may be indirectly assessed by blood lactate concentration, determining the capacity to use glucose as a fuel during high intensity efforts [66]. In this sense, higher glycolytic capacity, as found in this study and reported previously [66], can be translated into a better lactate tolerance and, therefore, a greater work capacity that allows athletes to perform longer at higher intensities. As described previously, the glycolytic pathway is directly regulated by glycogen content and glucose availability and, thus, lower glycogen and glucose disposal might limit this capacity [67]. It has been demonstrated that CHO intake during exercise could delay glycogen content and maintain blood glucose [54,68] and, hence, maintain greater high intensity run capacity. Therefore, the results obtained in this study could be explained by these mechanisms.

As glycogen represents a major regulator of glycolysis and, therefore, performance capacity [42,67], its resynthesis is a priority for athletes when constant efforts are required, such as multi-stage competitions or daily training sessions [25,28]. Thus, although glycogen resynthesis following exercise is a determining factor in ensuring suitable recovery and day-to-day performance, it is limited by muscle damage [25,69,70]. Therefore, lower EIMD could optimize glycogen resynthesis and shorten the recovery period following exercise. In this sense, we have recently demonstrated that 120 g/h CHO intake during exercise could limit muscle damage biomarkers 24 h following a mountain marathon [15]. This suggests that CHO intake during exercise could delay internal fatigue and limit EIMD and, thus, improve post- exercise glycogen replenishment and recovery. The results obtained in the present study have shown that, 24 h after completing a mountain marathon with ingestion of 120 g/h CHO, the time in terms of fatigue and blood lactate concentrations in glycolytic intensity (aerobic power-capacity test) was longer compared to current recommendations for ultra-endurance events (90 g/h) [34,35] and regular CHO intake by ultra-endurance runners (60 g/h) [36]. These findings led us to understand that a higher high intensity run capacity is maintained 24 h post-exercise by ingestion of a greater amount of CHO during the race, possibly due to a better glycogen content and, at the same point, because of less muscle damage. Overall, these results might confirm that CHO intake during exercise could represent not only a suitable strategy for enhancing performance as described in the literature [12,71], but also for optimizing long-term recovery and, thus, maintaining exercise performance in multi-stage competitions.

### 4.1. Limitations, Strengths and Future Lines of Research

This study presents some methodological limitations that could have improved the quality of the conclusions drawn. First of all, the activity was not carried out in a laboratory and, therefore, some relevant physiological variables such as oxygen consumption and substrate oxidation were not measured. As these could add important information about the athlete’s physiological and metabolic intensity during exercise, the authors assume this to be a major limitation. Moreover, the absence of glycogen measurements might represent a major limitation that could have enabled us to understand the mechanisms of and reasons for the findings [72]. A direct assessment of both central and peripheral neuromuscular function, was not carried out using specific tools such as electrical stimulation and electromyographic recordings, as done previously in this sports discipline [3,18,58]. On the other hand, using the same speed for all participants during the high intensity run capacity (20 km/h) could be a limitation. However, all the runners carried out the test between 1 and 3 min in both periods, which the interindividual variations allow to interpret individual fatigue. Lastly, although the athletes were instructed to extract the maximum content of the gels, a small part of this content could remain in the gel wrapper, which meant that the effective amount ingested by the athletes could be somewhat less than the theoretical amount.

Additionally, the study shows some strengths that justify its practical importance in the current literature. It was fully completed under real conditions with a mountain race organized by the authors. This involved a realistic effort from athletes and a unique opportunity for researchers to look into the real effects of CHO intake in a mountain marathon. In this sense, the fact that this trial was carried out in a real scenario helps us to understand the real effects of different CHO intakes in performance and recovery. As demonstrated in our previous research [15] this trial has allowed us to understand that 120 g/h CHO could be well tolerated in a physiologically highly demanding competition following suitable nutritional and gut training. This could prove to be highly relevant, as it contributes towards additional evidence provided by previous laboratory work [9,14,37].

These findings might open up some relevant future lines of research about the potential benefits of ingesting CHO during exercise to delay neuromuscular fatigue and high intensity run capacity, thus improving post-exercise recovery. Thus, more research is needed in order to understand the best recovery strategies in the field of sports and, specifically, in those multi-stage events in which recovery time is limited and plays a major role in performance. Moreover, this study, besides the previous one published by our group [15] showed that, within a practical and physiologically challenging scenario, the intake of 120 g/h CHO could be possible without resulting in any serious gastrointestinal problems, suggesting that current science-based recommendations of 90 g/h CHO for endurance events lasting more than 2.5 h could be subject to a rethink [11,35,71]. Accordingly, new lines of research could be opened up in order to understand which CHO ratio could be considered optimal in order to attain high amounts of ingestion. Moreover, as digestion, absorption and metabolic mechanisms are not well understood yet, future research is needed to move in line with scientific evidence. Lastly, this study highlights the need to (I) train the gut for endurance athletes to improve CHO intake, digestion, absorption and utilization during exercise and (II) research into the effects of training the gut to understand the mechanisms of these potential adaptations.

### 4.2. Practical Applications

This study underscores the importance of CHO ingestion during highly demanding endurance events in order to delay fatigue and improve recovery. Athletes competing in endurance and ultra-endurance events should train in nutritional planning and carry out gut training protocols to optimize performance and recovery from competitions. Moreover, coaches and nutritionists could use high CHO intake during exercise to improve high intensity run capacity and post-exercise recovery in multi-stage competitions or during extreme load training programs. It is our understanding that every athlete should focus on working on the individual maximum CHO intake possible (up to 120 g/h) to ingest without experiencing any gastrointestinal problems. Lastly, 2:1 glucose–fructose ratio could be considered a suitable composition for attaining high amounts of CHO ingestion, although more research is needed in this area.

## 5. Conclusions

An intake of 120 g/h CHO during a mountain marathon could limit neuromuscular fatigue. In addition, 120 g/h CHO during a mountain marathon would seem to improve long-term muscle recovery by limiting the decrease in neuromuscular function and high intensity run capacity 24 h following a challenging race compared to current recommendations for ultra-endurance events (90 g/h CHO) and regular intake by these athletes (60 g/h CHO). Therefore, high CHO intakes during highly demanding exercises of up to 120 g/h could represent a new, more suitable strategy for optimizing post-exercise recovery during consecutive efforts. Moreover, it has been demonstrated that 120 g/h CHO might be well tolerated in physiologically extreme conditions following a gut training plan.

## Figures and Tables

**Figure 1 nutrients-12-02094-f001:**
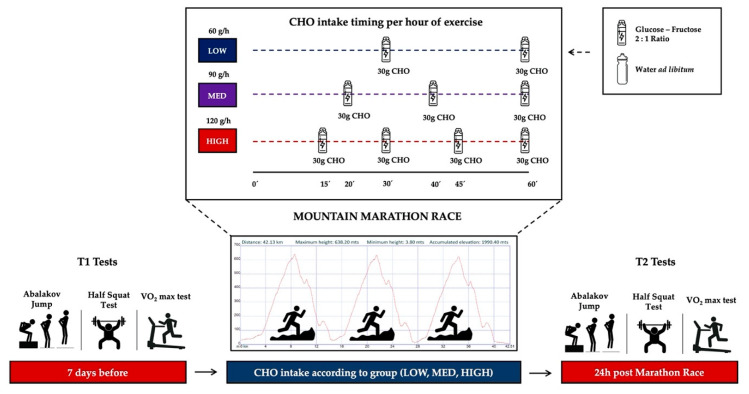
Experimental design of study including pre- and post- tests and race fueling protocol and timing. CHO: Carbohydrate.

**Figure 2 nutrients-12-02094-f002:**
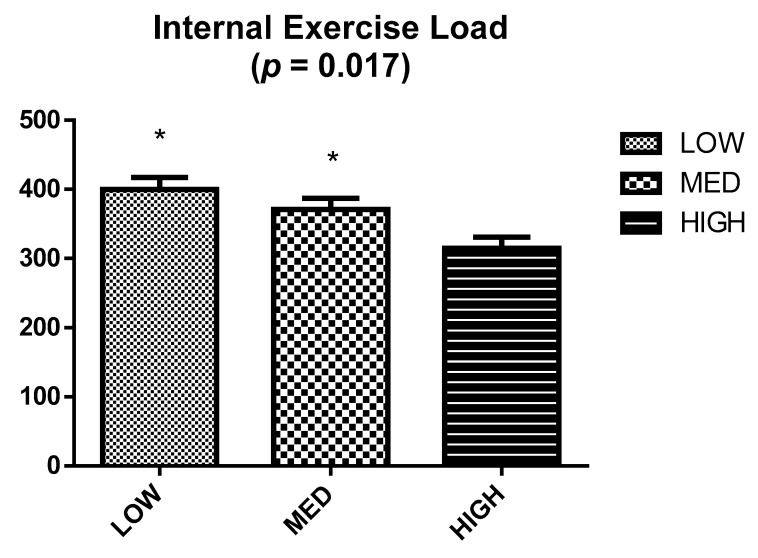
Internal exercise load by training impulse (TRIMP) for each group during trail marathon. * Significant differences in terms of HIGH using Mann Whitney *u* test (*p* < 0.05).

**Figure 3 nutrients-12-02094-f003:**
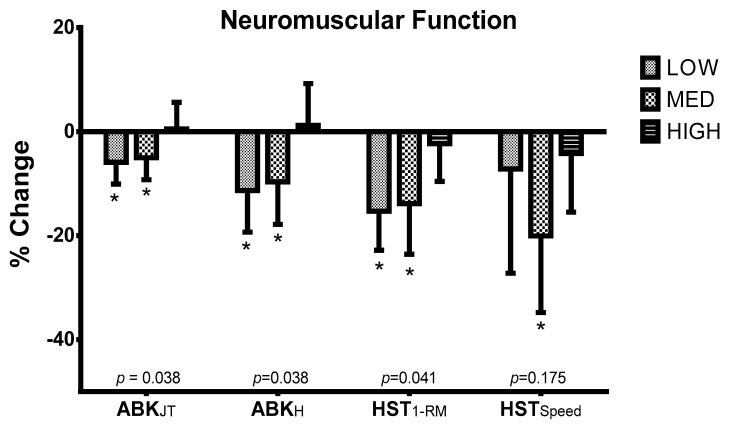
Percentage of changes of neuromuscular function using the Abalakov jump test and half squat test between T1 and T2. * Significant differences compared to HIGH using Mann Whitney *u* test (*p* < 0.05).

**Figure 4 nutrients-12-02094-f004:**
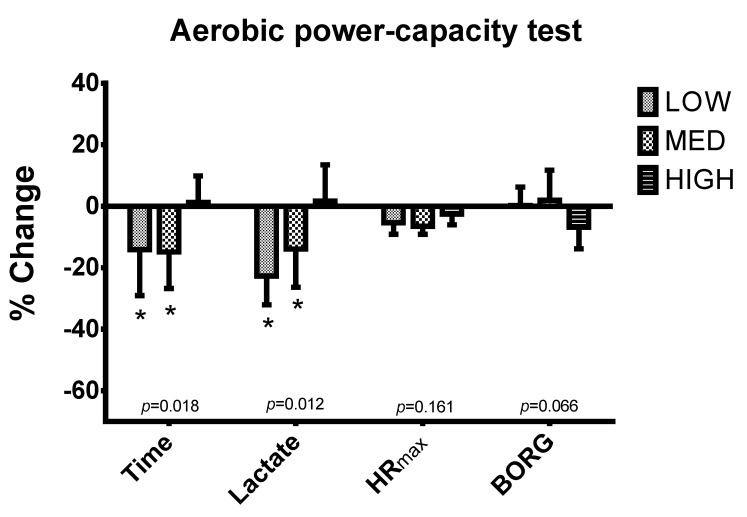
Percentage difference of high intensity run capacity parameters between T1 and T2. * Significant differences compared to HIGH using Mann Whitney *u* test (*p* < 0.05).

**Table 1 nutrients-12-02094-t001:** Race time, age, anthropometric characteristics and body composition in low (LOW), medium (MED) and high (HIGH) groups at baseline (T1).

Variables	LOW	MED	HIGH	*p*
Race Time (min)	278.2 ± 43.6	284.1 ± 40.0	271.7 ± 41.7	0.063
Age (years)	37.8 ± 9.4	37.2 ± 5.4	38.0 ± 6.8	0.639
Height (cm)	175.6 ± 10.3	172.3 ± 7.0	174.2 ± 3.5	0.361
Weight (kg)	71.8 ± 10.3	66.6 ± 10.1	67.4 ± 11.1	0.607
BMI	23.3 ± 2.9	22.4 ± 2.6	22.1 ± 3.0	0.747
∑6S (mm)	58.8 ± 21.5	55.4 ± 21.6	43.7 ± 21.6	0.467
Muscle Mass (kg)	29.8 ± 4.7	28.4 ± 5.1	30.4 ± 3.2	0.412

**Table 2 nutrients-12-02094-t002:** Results of the Abalakov jump and half-squad test in low (LOW), medium (MED) and high (HIGH) groups at T1 and T2.

Study Time	LOW	MED	HIGH	*p*
ABK_JT_ (s)				
T1	0.54 ± 0.05	0.53 ± 0.06	0.53 ± 0.05	0.867
T2	0.51 ± 0.05 *	0.50 ± 0.04 *	0.53 ± 0.04	0.867
ABK_H_ (cm)				
T1	36.57 ± 6.36	34.86 ± 7.37	34.46 ± 6.55	0.861
T2	32.42 ± 6.52 *	31.06 ± 4.97 *	34.47 ± 4.78	0.584
HST_1-RM_ (kg)				
T1	103.32 ± 35.67	109.07 ± 33.62	97.17 ± 8.60	0.991
T2	81.66 ± 32.42	91.91 ± 25.54	90.09 ± 14.20	0.659
HST_Speed_ (m/s)				
T1	0.65 ± 0.06	0.63 ± 0.10	0.57 ± 0.14	0.728
T2	0.60 ± 0.11	0.50 ± 0.09 *	0.55 ± 0.18	0.370

Data are indicated as mean ± standard deviation. *p*: Statistical differences among groups in each time point by Kruskal–Wallis test. * Significant differences (*p* < 0.05) between time points (T1 vs. T2) within the same group as determined by Wilcoxon signed rank test.

**Table 3 nutrients-12-02094-t003:** Results of the aerobic power-capacity test in low (LOW), medium (MED) and high (HIGH) groups before (T1) and after (T2) completing the competition.

Study Time	LOW	MED	HIGH	*p*
Time (s)				
T1	102.8 ± 38.3	104.0 ± 48.1	108.0 ± 46.5	0.962
T2	89.8 ± 37.1	87.9 ± 39.4 *	110.1 ± 48.4	0.537
Lactate (mmol/L)				
T1	7.45 ± 1.42	5.80 ± 0.91	6.79 ± 2.30	0.200
T2	5.65 ± 1.27 *	4.79 ± 1.35	6.70 ± 2.07	0.131
HR max (bpm)				
T1	184.8 ± 14.2	186.0 ± 11.0	179.6 ± 9.0	0.791
T2	174.7 ± 11.0 *	173.7 ± 8.1 *	174.9 ± 7.7	0.970
BORG				
T1	18.83 ± 1.17	18.43 ± 1.40	18.29 ± 0.76	0.008
T2	18.83 ± 0.98 ^&^	18.71 ± 1.38 ^&^	17.00 ± 1.00 *	0.028

Data are indicated as mean ± standard deviation. *p*: Statistical differences among groups in each time point by Kruskal–Wallis test. * Significant differences (*p* < 0.05) between time points (T1 vs. T2) within the same group as determined by Wilcoxon signed rank. ^&^ Significant differences regarding HIGH group by Mann Whitney *u* test.

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
