# Peer review of "Effects of 120 vs. 60 and 90 g/h Carbohydrate Intake during a Trail Marathon on Neuromuscular Function and High Intensity Run Capacity Recovery"

_nutrients, 2020, doi:10.3390/nu12072094_

Round 1

Reviewer 1 Report

Major

  • It is a very interesting study with rigorous design especially in controlling for the diet and between groups. Have authors collected blood so as to measure glucose, insulin levels, or some cytokine markers?
  • In the introduction and methods, please add more rationales regarding why a higher amount of CHO was given (120g/h). As more CHO was added during the race, more calories were added and some intolerance may also occur.
  • 6 Line 243: Your sample size per group was too small to conduct the one way ANOVA. You could use Krustal-Wallis to compare groups and then use Mann Whitney U tests to compare each two.
  • 6 Line 246: The two-way repeated measures ANOVA should have addressed interaction effect, including comparing T1 and T2 in each CHO group and comparing group difference at each time point so it is not necessary to conduct paired t test again. Please confirm.
  • Results, Table 2 – Lines 258-268: Because of the small sample size, the repeated measures ANOVA may not be appropriate. Typically, if the overall model on the interaction effect or main effect (group effect or time effect) was not significant, it is not meaningful to report if there were any differences between T1 and T2 (line 261). Did you see the time effect regardless of groups? I recommend that you run nonparametric using Wilcoxon signed rank test to compare T1 and T2 in each group, and use Krustal-Wallis to compare groups in each time point and then use Mann Whitney U tests to compare each two groups in each time point.

Minor

  • Line 116: change “31” to “Thirty-one”
  • Line 169: Was the personalize diet given 48 hours of period prior to T1 and marathon race day instead of T2?
  • Line 173: should delete “at T2” as this is not for T2, right?
  • Line 236: Please show Lee’s equation in the manuscript.
  • Lines 271-280: The font size should be adjusted.
  • Line 274: should be CON, not EXP

Author Response

Point-by-Point Response to Reviewer’s Comments

We would like to sincerely thank the reviewers for their helpful recommendations. We have seriously considered all the comments and carefully revised the manuscript accordingly. Revisions are highlighted in yellow through the manuscript to indicate where changes have taken place. We feel that the quality of the manuscript has been significantly improved with these modifications and improvements based on the reviewers’ suggestions and comments. We hope our revision will lead to an acceptance of our manuscript for publication in Nutrients.

In advance,

Kind regards

REVIEWER 1

Major

REVIEWER: It is a very interesting study with rigorous design especially in controlling for the diet and between groups. Have authors collected blood so as to measure glucose, insulin levels, or some cytokine markers?

AUTHORS: Thank you for your interest. We recently have published other manuscript about this research. This contained information about exercise induced muscle damage parameters like CK, LDH, GOT, urea, creatinine. Moreover, we obtained data of glucose. However, we did not measure insulin levels or cytokine marker, as explained in the limitations. In the future we will keep serum to could measure other parameters when will be necessary.

REVIEWER: In the introduction and methods, please add more rationales regarding why a higher amount of CHO was given (120g/h). As more CHO was added during the race, more calories were added and some intolerance may also occur more rationales regarding why a higher amount of CHO was given (120g/h):

AUTHORS: Thank you for your interest. The authors have included some sentences along the introduction and material and methods sections:

Line 61: “However, recent studies have shown that a gut training avoids gastrointestinal discomfort that it could facilitate the intake of greater amounts of CHO to the recommendations [15].

Line 76: In this sense, although >90 g/h CHO intake have controversial results [9,20], Pfeiffer et al. showed that athletes who consumed 120 g / h were among the fastest during 2 ultraendurance events, indicating a delay in the onset of fatigue [10]. Moreover, a study carried out by our lab has recently demonstrated that higher CHO intake (120 g/h) than recommended could be a determining factor in the internal exercise load response and could limit exercise-induced muscle damage (EIMD) in elite trail runners 24h after completing a mountain marathon, suggesting that recovery time after such an endurance event could be shortened by a suitable CHO intake during exercise [5].

Line 127: The use of 120 g / h was determined because previous studies have shown that after a training of gut tract it is possible to tolerate this amount [10,33] and that this amount of CHO during an ultraendurance race has shown a lower internal exercise load and lower EIMD [21].

REVIEWER: 6 Line 243: Your sample size per group was too small to conduct the one way ANOVA. You could use Krustal-Wallis to compare groups and then use Mann Whitney U tests to compare each two.

AUTHORS: Thank you for your recommendation. The authors have remade the tables and figures based on the new statistics test. Likewise, the authors have changed the Statistical Data Analyses section: “Although the data obtained presented a parametric distribution after using the Shapiro-Wilk test (n <50), non-parametric tests were performed because the sample in each of the study group was very small. The percentage changes of the physical test between T1 and T2 were considered as Δ (%): [(T2 – T1) / T1] × 100. Δ (%) of TRIMP and athletic recovery (ABK, DJ, HST and VO2 max test) was compared among 3 CHO intake groups using Kruskal-Wallis test with the CHO intake groups as the fixed factor. A Mann Whitney U tests test was completed for pairwise comparisons between groups.

Similarly, differences between T1 and T2 in each athletic recovery test in each CHO intake group were assessed using a Wilcoxon signed rank test.”

REVIEWER: 6 Line 246: The two-way repeated measures ANOVA should have addressed interaction effect, including comparing T1 and T2 in each CHO group and comparing group difference at each time point so it is not necessary to conduct paired t test again. Please confirm.

AUTHORS: Thank you for your observation. Following reviewer´s recommendation in the next point the authors have deleted the two-way repeated measures ANOVA and we have used Wilcoxon signed rank test to compare T1 and T2 in each group, and use Krustal-Wallis to compare groups in each time point and then use Mann Whitney U tests to compare each two groups in each time point.

REVIEWER: Results, Table 2 – Lines 258-268: Because of the small sample size, the repeated measures ANOVA may not be appropriate. Typically, if the overall model on the interaction effect or main effect (group effect or time effect) was not significant, it is not meaningful to report if there were any differences between T1 and T2 (line 261). Did you see the time effect regardless of groups? I recommend that you run nonparametric using Wilcoxon signed rank test to compare T1 and T2 in each group, and use Krustal-Wallis to compare groups in each time point and then use Mann Whitney U tests to compare each two groups in each time point.

AUTHORS: Thank you for your recommendation. As we have previously indicated the authors have remade the tables and figures based on the new statistics test. Likewise, the authors have changed the Statistical Data Analyses section.

Minor

REVIEWER: Line 116: change “31” to “Thirty-one”

AUTHORS: Thank you, we have changed “31” to “Thirty-one”

REVIEWER: Line 169: Was the personalize diet given 48 hours of period prior to T1 and marathon race day instead of T2?

AUTHORS: Thank you for your observation. The authors have changed T2 to marathon race day:The athletes obtained a personalized diet for a 48-hour period prior to T1 and marathon race day with 9 g CHO/kg BM/day, 1.5 g protein/kg BM/day and 0.5 g fat/kg BM/day to adjust their glycogen content.”

REVIEWER: Line 173: should delete “at T2” as this is not for T2, right?

AUTHORS: Thank you. We have deleted “at T2”.

REVIEWER: Line 236: Please show Lee’s equation in the manuscript.

AUTHORS: Thank you for your recommendation. The authors have included Lee´s equation in Anthropometry and Body Composition section:A Lufkin® model W606PM measuring tape with 1 mm precision was used to measure the perimeters (relaxed arm, mid-thigh and calf) in cm. These girths were all corrected for the skinfold at the site using the following formula: [corrected girth = girth – (π x skinfold thickness at the site)]. Relaxed arm girth was corrected for triceps skinfold (CAG), mid-thigh girth corrected for front thigh skinfold (CTG), and calf girth was corrected for medial calf skinfold (CCG). Muscle mass (MM) was predicted using the Lee [6] equation for males and Caucasian athletes:

MM = height × (0.00744 × CAG2 + 0.00088 × CTG2 + 0.00441 × CCG2) + 2.4 − 0.048 × age + 7.8”

REVIEWER: Lines 271-280: The font size should be adjusted.

AUTHORS: Thank you for your observation, the font size has been adjusted.

REVIEWER: Line 274: should be CON, not EXP

AUTHORS: Thank you for your observation, we have changed EXP to CON.

Reviewer 2 Report

The present manuscript outlines the effects of three different carbohydrate intake rates across an off-road marathon event, on recovery of high intensity run capacity, a series of explosive jump tests, and a squad-based strength test based on velocity monitoring of a submaximal loads. The authors also explore the impact of different CHO intakes on internal training load assessed by HR monitoring and subsequent TRIMP method. The study design is very appropriate and more applied research of this nature is required to understand and advance our field. The dietary provision and prescription appear very well controlled. There are, however, some concerns about the actual intake compared to prescribed intake, as this can often vary despite the best prepared plans. I also have some concern about the testing and statistical methods employed. Finally, the discussion section needs much more focus on the data found in this study. Currently, it reads more like a literature review, and several paragraphs in a row barely touch on the data from this investigation. This section needs significant work. Nonetheless, I think the data is interesting and with revisions this manuscript could be of much interest.

Major comments:

The introduction needs to be streamlined and more focused. The authors demonstrate a good understanding of the current literature in the area of CHO metabolism, but the introduction keeps moving in different directions and doesn’t flow logically to the aims of this study. I think maybe there needs to be some restructure, and minor comments below could assist with this.

How did you verify all CHO was consumed by each participant in each group? How were gels prepared/ stored for consumption? Not every gram of a gel can be squeezed out of its packaging during exercise! As such, how do you know exactly how much CHO was consumed? If there’s no mechanism in place to ensure/measure consumption, this needs to be reported as a limitation.

Can you please remove as many non-standard abbreviations throughout the manuscript as possible? I’m certainly happy with abbreviations like CHO and VO2peak that the reader in this field will be familiar with, but HST, and AKB for Abalakov jump test doesn’t make sense, and hinders readability.

The range of statistical tests you have here is not easy to follow or understand why you have done these analyses? The first delta% analyses doesn’t make sense to me at all. A repeated measures ANOVA with three groups, and two timepoints should be used only (as you state below). If this shows nothing (no interaction, or main effects), that’s fine. You follow this up with within-group t-tests anyhow. I also wonder why you have used partial squared ES? Can these be replaced (at least for pre-post comparisons) with a more common ES like Cohen’s d? If there’s not a very good rationale for the delta% analysis, this needs to be removed please.

Related to the statistical analyses, line 259 states a significant decrease was noted… using which analyses? Please explicitly state this. In this regard, can you please be more methodical in how you report each section of your results. For example, first state if there was/wasn’t an interaction or main effect. Next, move to the within-group analyse, p-value (probability test) and Cohen’s d ES (magnitude based). Having this consistency really helps the reader. Also, from line 263 (and others), please remove the data from in-text – it’s displayed well in the tables directly below. Currently, it’s quite difficult to read the results section as it’s cluttered with replicated data. Finally, if there was a ‘tendency’ towards a smaller loss, can you state the p-value/effect size at each instance? Please use exact p-values here and throughout. If these are the within-group changes, also show the Cohens d ES to demonstrate how big the change was. 

The discussion section needs a complete restructure. The initial paragraph is well written and outlines the main findings, but thereafter these findings are barely mentioned. For example, the paragraph starting line 353 is written entirely without any reference to your data. Please present a point of data for discussion up front, and write the paragraph around that point/data. The same issues is evident in the paragraph starting line 375, where there is no reference to your data until line 384. Again, the same for paragraph starting 390 where you discuss the role of glycogen, but you do not measure glycogen, nor any measure relating to metabolic flexibility. This is a common trend (discussion paragraphs not related to you data!) that I will leave you to attend. The discussion needs to be much more focussed.  

Minor comments:

  • line 21: exercise load? internal load assessed by? There is too little information here for anyone not familiar with this prior paper. Please either remove or provide further information.
  • line 22: remove: “This suggests that CHO intake during exercise might optimize recovery” as it’s not required here.
  • line 25-26 glycolytic function recovery is ambiguous, and I have never come across this wording. Please replace recovery of high intensity run capacity, or similar (here and throughout the paper).
  • line 27: could you not name your groups low med/mod and high? I understand the rationale for the current naming system based on your introduction, but many readers may be confused as to why the control is the ‘recommended’ rather than the ‘actual’ intake that runners consume. To remove this issue, I suggest the above naming convention.
  • line 30: state how many days after the race here.
  • line 31: what are these subscript abbreviations? They’re not defined anywhere. These cannot be used unless defined. I appreciate the word restrictions in an abstract, but it’s impossible for the reader to understand in its current form.
  • 31-32: given there’s no mention of statistical analysis here (which I’m fine with for an abstract), it’s not clear what these statistical differences refer to? Are these only pre-post within-group changes? I assume they’re not posthoc tests from a 2x3 RM ANOVA? This is important to note either way.
  • 34-35: which way were these differences, increased/decreased x, y and z?
  • please remove the last sentence of abstract – this was not an outcome of your study.
  • line 42: maybe rearrange, as this sentence sets the scene and could flow better. For example: The different distances run by participants range from mountain marathons (42,195 m) to multistage ultra-marathons (up to 350km), with an accumulative altitude gain of ~24,000 m during the most extreme events.
  • line 46 environmental conditions?
  • line 53: monitoring perceptions of fatigue? I am still not quite sure what the ‘internal load’ you are referring to is? My view of an ‘internal load’ is something like a session RPE or TRIMP which can be summed across time/sessions but represents the ‘load’ (intensity [measured or perceived] x time) of a session. If we are referring to perceptions of soreness, fatigue, etc, I wouldn’t suggest this is a load? I’m also not sure how an internal load gives us specific insight to neuromuscular fatigue? This needs to be cleared up please.
  • line 55: this paragraph is opened by starting CHO intake delays fatigue (improves exercise capacity?) and improves performance (improved work rate?) in a dose-response manner – but there is no mention of the performance aspect in this paragraph. As mentioned under major comments above, the introduction needs to be more focussed and a logical flow, rather than jumping between ideas. Maybe keep focused on the link between CHO intake and exercise capacity/delay in fatigue? More applied studies that demonstrate improved capacity. After this, focus on the mechanisms of fatigue in the next paragraph by use of Scandinavian studies?
  • line 63: over what time frame is this supplementation? In what population? Doesn't add much meaning in it's currently written format.
  • line 66: suggest opening this paragraph by stating something on the mechanisms responsible for peripheral fatigue. Several of your paragraphs are introduced very passively (“on the other hand”, “Additionally”). Can these be amended please?
  • line 70-74: maybe just focus on the post-ex 4h time-point? Is the rationale because of less depletion during exercise – it’s currently unclear? I don’t think there is need to go into sub-cellular fractions given the applied nature of your work (and that no glycogen measure took place), maybe just a rationale for why you believe there is benefit to higher CHO intakes.
  • line 79-85: I’m not sure how this section relates to your study? There is no intervention regarding the provision of CHO after exercise? same for 91-93. This could be removed, and maybe focus this paragraph on the time-course of recovery?
  • line 96-98: maybe soften this statement? What about using protein during exercise to show a more positive protein balance during simulated ultra -exercise? (Koopman et al. 2007 Am J Physiol Endo Metab 287).
  • line 112: can you name the marathon here in the intro/overview of the study design?
  • line 120: why are people who have done gut training excluded?
  • Line 116 and 128: can you between describe you participants here, or move table 1 here? You are describing the participants in the trials, but there’s no data on them until the results section, and I’d argue these aren’t results as such. Can you also display baseline VO2peak here? Some indication of training status of the entire group.
  • line 143-149: can this be moved to participants section above? why does the reader have to wait until here to now that only 20 completed the study!
  • can HRM be HRmean for consistency, given HRmax?
  • line 192: Can you explain the hands free jump? Have these tests been previously used to measure NMF? what is the reliability of measures?
  • 198: there’s no mention of DJ test yet? What is this?
  • I’m confused here. Did the runners actually do a 1RM at any point? How was 70% calculated? This section is not quite and requires clarification please.
  • line 208: this variable name is too specific for what is actually being measure here. Maybe high intensity exercise/run capacity. Also, why 20km/h? Is this the same %max for all participants? Why did you not individualise this for participants? Seems like a limitation that needs mentioning. Finally, is time to fatigue the key outcome here? Was VO2 measured at all? If not, the variable description certainly must change.
  • 223: reference for this adjustment factor? This method needs to be better described, in full please as the equation is also difficult to follow.
  • line 242: what is the change in TRIMP? Wasn’t the load only obtained once during the race?
  • line 271: what is the CV capacity test – this is the first time this has been referred to as this.
  • line 289: compared to EXP – please amend here and throughout.
  • figure 3 and 4 – as of above major comments, I think the stats in which these are based are not well justified. With the restructure proposed, I will leave you to decide if these stay or go. If you have good justification for them/the analysis, please alter the key in these figures as I cannot tell which is LOW and which is EXP.
  • line 388: I think this is a stretch to claim they were faster. I would focus only on the recovery/TRIMP which I believe is a nice finding – can you expanded on this specifically, and why this has occurred?

Author Response

Point-by-Point Response to Reviewer’s Comments

We would like to sincerely thank the reviewers for their helpful recommendations. We have seriously considered all the comments and carefully revised the manuscript accordingly. Revisions are highlighted in yellow through the manuscript to indicate where changes have taken place. We feel that the quality of the manuscript has been significantly improved with these modifications and improvements based on the reviewers’ suggestions and comments. We hope our revision will lead to an acceptance of our manuscript for publication in Nutrients.

In advance,

Kind regards

REVIEWER 2

The present manuscript outlines the effects of three different carbohydrate intake rates across an off-road marathon event, on recovery of high intensity run capacity, a series of explosive jump tests, and a squad-based strength test based on velocity monitoring of a submaximal loads. The authors also explore the impact of different CHO intakes on internal training load assessed by HR monitoring and subsequent TRIMP method. The study design is very appropriate and more applied research of this nature is required to understand and advance our field. The dietary provision and prescription appear very well controlled. There are, however, some concerns about the actual intake compared to prescribed intake, as this can often vary despite the best prepared plans. I also have some concern about the testing and statistical methods employed. Finally, the discussion section needs much more focus on the data found in this study. Currently, it reads more like a literature review, and several paragraphs in a row barely touch on the data from this investigation. This section needs significant work. Nonetheless, I think the data is interesting and with revisions this manuscript could be of much interest.

Major comments:

REVIEWER: The introduction needs to be streamlined and more focused. The authors demonstrate a good understanding of the current literature in the area of CHO metabolism, but the introduction keeps moving in different directions and doesn’t flow logically to the aims of this study. I think maybe there needs to be some restructure, and minor comments below could assist with this.

AUTHORS: Thank you for your recommendations. The authors have tried improve this point answering minor comments.

REVIEWER: How did you verify all CHO was consumed by each participant in each group? How were gels prepared/ stored for consumption? Not every gram of a gel can be squeezed out of its packaging during exercise! As such, how do you know exactly how much CHO was consumed? If there’s no mechanism in place to ensure/measure consumption, this needs to be reported as a limitation.

AUTHORS: Thank you for your observations. Regarding how we verified all athletes consumed gels, we have added next sentence in Participants and experimental protocol section: “The athletes were instructed to extract the maximum content of the gels so that the residual amount was minimal. Likewise, all the athletes who finished the race confirmed through a questionnaire that all the gels had been taken at the time and in the manner previously indicated.”

Respect to how gels were prepared, we have included this sentence in Participants and experimental protocol section: All participants ingested CHO during the race using the same gels made for this research at the University of Valladolid Physiology Laboratory (Soria) by an experienced pharmacist using a gel packaging machine.

Lastly, although the athletes were instructed to extract the maximum content of each gel, we are aware that a small part could be left in the gel wrapper. In this sense, the authors have included a phrase in the limitations section: “Lastly, although the athletes were instructed to extract the maximum content of the gels, a small part of this content could remain in the gel wrapper, which meant that the effective amount ingested by the athletes could be somewhat less than the theoretical amount.”

REVIEWER: Can you please remove as many non-standard abbreviations throughout the manuscript as possible? I’m certainly happy with abbreviations like CHO and VO2peak that the reader in this field will be familiar with, but HST, and AKB for Abalakov jump test doesn’t make sense, and hinders readability.

AUTHORS: Thank you for your recommendation. The authors have changed HST and ABK for half-squad test and Abalakov jump test respectively.

REVIEWER: The range of statistical tests you have here is not easy to follow or understand why you have done these analyses? The first delta% analyses doesn’t make sense to me at all. A repeated measures ANOVA with three groups, and two timepoints should be used only (as you state below). If this shows nothing (no interaction, or main effects), that’s fine. You follow this up with within-group t-tests anyhow. I also wonder why you have used partial squared ES? Can these be replaced (at least for pre-post comparisons) with a more common ES like Cohen’s d? If there’s not a very good rationale for the delta% analysis, this needs to be removed please.

AUTHORS: Thank you for your recommendation. Following your and reviewer 1 comments (“I recommend that you run nonparametric using Wilcoxon signed rank test to compare T1 and T2 in each group, and use Krustal-Wallis to compare groups in each time point and then use Mann Whitney U tests to compare each two groups in each time point.”), the authors have remade the tables and figures based on the new statistics test. Likewise, the authors have changed the Statistical Data Analyses section:

“Statistical analysis was completed by Statistical Package for the Social Sciences 24.0 (SPSS Inc. Chicago, IL, USA), with results being shown as mean and standard deviation. The significance level for all analyses was set at p < 0.05.

Although the data obtained presented a parametric distribution after using the Shapiro-Wilk test (n <50), non-parametric tests were performed because the sample in each of the study group was very small. The percentage changes of the physical test between T1 and T2 were considered as Δ (%): [(T2 – T1) / T1] × 100. Δ (%) of TRIMP and athletic recovery (ABK, DJ, HST and VO2 max test) was compared among 3 CHO intake groups using Kruskal-Wallis test with the CHO intake groups as the fixed factor. A Mann Whitney U tests test was completed for pairwise comparisons between groups.

Similarly, differences between T1 and T2 in each athletic recovery test in each CHO intake group were assessed using a Wilcoxon signed rank test.”

On the other hand, regarding delta% analyses we believe that is very important to show the % differences. We have done this analysis in a multitude of published manuscripts so that the reader can clearly see what the percentage change was during the study. Tables 2 and 3 show the absolute values in each group in T1 and T2. Do not hesitate, if the reviewer insists that we should delete these data, we would be happy to do so in the next round of reviews.

REVIEWER: Related to the statistical analyses, line 259 states a significant decrease was noted… using which analyses? Please explicitly state this. In this regard, can you please be more methodical in how you report each section of your results. For example, first state if there was/wasn’t an interaction or main effect. Next, move to the within-group analyse, p-value (probability test) and Cohen’s d ES (magnitude based). Having this consistency really helps the reader. Also, from line 263 (and others), please remove the data from in-text – it’s displayed well in the tables directly below. Currently, it’s quite difficult to read the results section as it’s cluttered with replicated data. Finally, if there was a ‘tendency’ towards a smaller loss, can you state the p-value/effect size at each instance? Please use exact p-values here and throughout. If these are the within-group changes, also show the Cohens d ES to demonstrate how big the change was. 

AUTHORS: Thank you for your observation. As we have said in the previous comment, we have modified the statistical analysis based on your comments and those of the reviewer 1. Although in general the statistical results are similar by a parametric route rather than a non-parametric one, this has led to the modification of the tables and therefore the results.

REVIEWER: The discussion section needs a complete restructure. The initial paragraph is well written and outlines the main findings, but thereafter these findings are barely mentioned. For example, the paragraph starting line 353 is written entirely without any reference to your data. Please present a point of data for discussion up front, and write the paragraph around that point/data. The same issues is evident in the paragraph starting line 375, where there is no reference to your data until line 384. Again, the same for paragraph starting 390 where you discuss the role of glycogen, but you do not measure glycogen, nor any measure relating to metabolic flexibility. This is a common trend (discussion paragraphs not related to you data!) that I will leave you to attend. The discussion needs to be much more focussed.

AUTHORS: Thank you for your correction. The authors agree with the reviewer and, thus, have modified the different paragraphs of the discussion following your suggestions.

Minor comments:

REVIEWER: line 21: exercise load? internal load assessed by? There is too little information here for anyone not familiar with this prior paper. Please either remove or provide further information.

AUTHORS: Thank you for your recommendation. The authors have deleted “exercise load”.

REVIEWER: line 22: remove: “This suggests that CHO intake during exercise might optimize recovery” as it’s not required here.

AUTHORS: Thank you for your recommendation. We have deleted this sentence.

REVIEWER: line 25-26 glycolytic function recovery is ambiguous, and I have never come across this wording. Please replace recovery of high intensity run capacity, or similar (here and throughout the paper).

AUTHORS: Thank you for your recommendation. The authors have changed glycolytic function for high intensity run capacity throughout the paper.

REVIEWER: line 27: could you not name your groups low med/mod and high? I understand the rationale for the current naming system based on your introduction, but many readers may be confused as to why the control is the ‘recommended’ rather than the ‘actual’ intake that runners consume. To remove this issue, I suggest the above naming convention.

AUTHORS: Thank you for your recommendation. We have changes CON for MED and EXP for HIGH throughout the paper.

REVIEWER: line 30: state how many days after the race here.

AUTHORS: Thank you for your observation. The authors have added “24 h after completing the race.”

REVIEWER: line 31: what are these subscript abbreviations? They’re not defined anywhere. These cannot be used unless defined. I appreciate the word restrictions in an abstract, but it’s impossible for the reader to understand in its current form.

AUTHORS: Thank you for your observation. We have added information about these abreviations:Changes in Abalakov jump time (ABKJT), Abalakov jump height (ABKH), half-squat test 1 repetition maximum (HST1RM) between T1 and T2 showed significant differences only in LOW and MED (p<0.05), but not in the HIGH group (p>0.05).”

REVIEWER: 31-32: given there’s no mention of statistical analysis here (which I’m fine with for an abstract), it’s not clear what these statistical differences refer to? Are these only pre-post within-group changes? I assume they’re not posthoc tests from a 2x3 RM ANOVA? This is important to note either way.

AUTHORS: Thank you for your interest. The authors have rewritten the results in abstract: Results: Changes in Abalakov jump time (ABKJT), Abalakov jump height (ABKH), half-squat test 1 repetition maximum (HST1RM) between T1 and T2 showed significant differences by Wilcoxon signed rank test only in LOW and MED (p<0.05), but not in the HIGH group (p>0.05). Internal load was significantly lower in the HIGH group (p=0.017) regarding LOW and MED by Mann Whitney U test. Significant lower change during the study in ABKJT (P=0.038), ABKH (P=0.038) HST1RM (P=0.041) and in terms of fatigue (p=0.018) and lactate (p=0.012) within the VO2 max test was presented in HIGH respect to LOW and MED.

REVIEWER: 34-35: which way were these differences, increased/decreased x, y and z?

AUTHORS: Thank you for your comment. This issue was corrected with the previous point.

REVIEWER: please remove the last sentence of abstract – this was not an outcome of your study.

AUTHORS: Thank you for your recommendation. That sentence was deleted from abstract.

REVIEWER: line 42: maybe rearrange, as this sentence sets the scene and could flow better. For example: The different distances run by participants range from mountain marathons (42,195 m) to multistage ultra-marathons (up to 350km), with an accumulative altitude gain of ~24,000 m during the most extreme events.

AUTHORS: Thank you for your recommendation. The authors have modified this sentence: The different distances run by participants range from mountain marathons (42,195 m) to multistage ultra-marathons (up to 350km), with an accumulative altitude gain of ~24,000 m during the most extreme events.”

REVIEWER: line 46 environmental conditions?

AUTHORS: Thank you for your observation. We have added environmental for external.

REVIEWER: line 53: monitoring perceptions of fatigue? I am still not quite sure what the ‘internal load’ you are referring to is? My view of an ‘internal load’ is something like a session RPE or TRIMP which can be summed across time/sessions but represents the ‘load’ (intensity [measured or perceived] x time) of a session. If we are referring to perceptions of soreness, fatigue, etc, I wouldn’t suggest this is a load? I’m also not sure how an internal load gives us specific insight to neuromuscular fatigue? This needs to be cleared up please.

AUTHORS: Thank you for your suggestion. The authors agree with the review and have clarified this part to avoid misunderstandings.

REVIEWER: line 55: this paragraph is opened by starting CHO intake delays fatigue (improves exercise capacity?) and improves performance (improved work rate?) in a dose-response manner – but there is no mention of the performance aspect in this paragraph. As mentioned under major comments above, the introduction needs to be more focused and a logical flow, rather than jumping between ideas. Maybe keep focused on the link between CHO intake and exercise capacity/delay in fatigue? More applied studies that demonstrate improved capacity. After this, focus on the mechanisms of fatigue in the next paragraph by use of Scandinavian studies?

AUTHORS: Thank you for your correction. The authors have reorganized the paragraph following a clearer structure:

Carbohydrate (CHO) intake during endurance exercise has been shown to delay neuromuscular fatigue and improve exercise capacity and work rate significantly in a dose-response relationship [9,10]. Current recommendations in events lasting more than 2.5 hours include the 90 g /h CHO intake, with a combination of multiple CHO to maximize intestinal absorption using different transporters (GLUT5 for fructose and SGLT-1 for glucose) [11,12]. Moreover, recent studies have shown that a gut training avoids gastrointestinal discomfort that it could facilitate the intake of greater amounts of CHO to the recommendations [15]. In this sense, although >90 g/h CHO intake have controversial results [9,20], Pfeiffer et al. showed that athletes who consumed 120 g / h were among the fastest during 2 ultraendurance events, indicating a delay in the onset of fatigue [10]. Moreover, a study carried out by our lab has recently demonstrated that higher CHO intake (120 g/h) than recommended could be a determining factor in the internal exercise load response and could limit exercise-induced muscle damage (EIMD) in elite trail runners 24h after completing a mountain marathon, suggesting that recovery time after such an endurance event could be shortened by a suitable CHO intake during exercise [21].  In addition, it is well known that CHO substrate availability plays a central role in peripheral fatigue, but also within the central nervous system and, thus, in central fatigue [16]. Stewart et al. [17] showed that glucose intake (1.23 ± 0.11 g/kg body mass) during exercise with a 15 min frequency in 15 untrained participants improved muscle function due to attenuated disturbances in the membrane excitability, suggesting that peripheral fatigue could be delayed by CHO intake during exercise.

REVIEWER: line 63: over what time frame is this supplementation? In what population? Doesn't add much meaning in it's currently written format.

AUTHORS: Thank you for your appreciation. The authors have added the information required:

Stewart et al. [17] showed that glucose intake (1.23 ± 0.11 g/kg body mass) during exercise with a 15 min frequency in 15 untrained participants improved muscle function due to attenuated disturbances in the membrane excitability, suggesting that peripheral fatigue could be delayed by CHO intake during exercise.

REVIEWER: line 66: suggest opening this paragraph by stating something on the mechanisms responsible for peripheral fatigue. Several of your paragraphs are introduced very passively (“on the other hand”, “Additionally”). Can these be amended please?

Peripheral fatigue is understood as the failure of local mechanisms in the muscle and, therefore, the decrease in the contraction and relaxation function that are related with the energetic status of the muscle cell [19,20]. Among these mechanisms, the potential action in the sarcolemma, the excitation-contraction (E-C) coupling and the interaction between actin and myosin proteins that allowed the muscle to contract and relax [21].

AUTHORS: Thank you for your suggestion. The authors have added this information.

REVIEWER: line 70-74: maybe just focus on the post-ex 4h time-point? Is the rationale because of less depletion during exercise – it’s currently unclear? I don’t think there is need to go into sub-cellular fractions given the applied nature of your work (and that no glycogen measure took place), maybe just a rationale for why you believe there is benefit to higher CHO intakes.

AUTHORS: Thank you for your appreciation. The authors have focused on the 4h recovery results as they represent the importance of this manuscript.

In this sense, the link between localized intramyofibrillar glycogen content and muscle function, mediated by the Ca2+ release from the sarcoplasmic reticulum (SR), has been established in literature [22,23]. A study conducted on elite cross-country skiers showed a correlation between the reduction in skeletal muscle glycogen and release rate of Ca2+ after completing 1h of maximum effort [24]. Moreover, significant differences were found in glycogen content and Ca2+ release after 4h post-exercise between the group that consumed CHO (1 g/kg body weight/h) and the placebo one (water), suggesting that CHO intake during exercise might take on a major role in improving short term glycogen replenishment and muscle function [24].

REVIEWER: line 79-85: I’m not sure how this section relates to your study? There is no intervention regarding the provision of CHO after exercise? same for 91-93. This could be removed, and maybe focus this paragraph on the time-course of recovery?

AUTHORS: Thank you for your recommendation. With the addition of information in previous paragraphs the authors consider important this paragraph for understanding the recovery kinetics and for clearing that the recovery of glycogen and neuromuscular function is not full after 24h. This is essential to understand why the strategy carried out during the race could be determinant for the recovery process.

REVIEWER: line 96-98: maybe soften this statement? What about using protein during exercise to show a more positive protein balance during simulated ultra -exercise? (Koopman et al. 2007 Am J Physiol Endo Metab 287).

AUTHORS: Thank you for your suggestion. The authors have softened this statement adding some references that used CHO and protein during exercise:

Although few studies have been conducted with the ingestion of protein and carbohydrates during the exercise with the aim of improving recovery [33–35], ingested CHO quantities were lower than currently recommended [12]. To the best of the authors’ knowledge, no research has been conducted using only CHO in high doses during exercise to optimize post-exercise recovery, which could prove very interesting when it comes to improving training capacity and performance in multi-stage competitions.

REVIEWER: line 112: can you name the marathon here in the intro/overview of the study design?

AUTHORS: Thank you for your recommendation. The name of marathon is marathon of Oiartzun: The “marathon of Oiartzun” is a mountain marathon race (42.195 km) which began at 9:00 am in Oiartzun (Guipúzcoa-Spain) (10 ºC, 60 % humidity and 10 km/h wind speed) and was controlled by official chronometers.

REVIEWER: line 120: why are people who have done gut training excluded?

AUTHORS: Thank you for your observation. The authors have included this sentence to clarify this point: “(>5 years’ experience in ultra-endurance training, having undertaken personalized training of the gut tract to enhance CHO absorption capacity and tolerance and not taken any drugs or performance supplements [34] to avoid any possible interference o in the recovery process during the 1-week period prior to the race).”

REVIEWER: Line 116 and 128: can you between describe you participants here, or move table 1 here? You are describing the participants in the trials, but there’s no data on them until the results section, and I’d argue these aren’t results as such. Can you also display baseline VO2peak here? Some indication of training status of the entire group.

AUTHORS:

REVIEWER: line 143-149: can this be moved to participants section above? why does the reader have to wait until here to now that only 20 completed the study!

AUTHORS: Thank you for your observation. That paragraph has been moved after the part where it talks about the randomization of athletes (line 138-145).

REVIEWER: can HRM be HRmean for consistency, given HRmax?

AUTHORS: Thank you for your comment. The authors have changed HRM be HRmean .

REVIEWER: line 192: Can you explain the hands free jump? Have these tests been previously used to measure NMF? what is the reliability of measures?

AUTHORS: Thank you for your comment. to avoid misunderstandings the authors have deleted “hands free jump to avoid.

On the other hand, the authors have included some references: “Abalakov jump test [46] and half squat [47] test were chosen to measure the neuromuscular function of leg extensor muscles in athletes and recreationally active men given that they can achieve this with a high degree of reliability.”

REVIEWER: 198: there’s no mention of DJ test yet? What is this?

AUTHORS: Thank you for your observation. DJ has been changed for Abalakov jump test.

REVIEWER: I’m confused here. Did the runners actually do a 1RM at any point? How was 70% calculated? This section is not quite and requires clarification please.

AUTHORS:  Thank you for your interest. This half-Squad-test was used to determinate 1 RM and concentric movement speed. In this sense, the authors have rewritten this section to clarify these two tests: “Half Squat test: Five-minutes following Abalakov jump test, runners performed a 1-repetition maximum test (1RM) of half squat test (HST1-RM) using a Multipower machine (BH Max Rack LD400, Vitoria, Spain). After 5 minutes’ rest from HST1-RM the runners performed three repetitions at maximal speed for a load of 70% of their 1-RM in this half-squad exercise, with 1-min rest between repetitions to determinate the concentric movement speed of half-squad test (HSTSpeed).”

REVIEWER: line 208: this variable name is too specific for what is actually being measure here. Maybe high intensity exercise/run capacity. Also, why 20km/h? Is this the same %max for all participants? Why did you not individualise this for participants? Seems like a limitation that needs mentioning. Finally, is time to fatigue the key outcome here? Was VO2 measured at all? If not, the variable description certainly must change.

AUTHORS: Thank you for your observation. Following your previous comments, we have changes glycolytic capacity for High intensity run capacity.

On the other hand, All the participants ran at 20 km / h. Although as the reviewer indicates it could be a limitation, none of the athletes managed to run for more than 3 minutes. Furthermore, in both moments of the study they ran at the same speed, so that the interindividual variations allow to interpret fatigue. In this sense, the authors have included this sentence in limitation section: “On the other hand, using the same speed for all participants during the high intensity run capacity (20 km/h) could be a limitation. However, all the runners carried out the test between 1 and 3 minutes in both periods, which the interindividual variations allow to interpret individual fatigue.”

Given that we have not measure VO2 max we have changed this term for aerobic power-capacity test.

REVIEWER: 223: reference for this adjustment factor? This method needs to be better described, in full please as the equation is also difficult to follow.

AUTHORS: Thank you for your observation. We have included the equation: ΔHR = (HRmean - resting HR)/ (HRmax - resting HR).

REVIEWER: line 242: what is the change in TRIMP? Wasn’t the load only obtained once during the race?

AUTHORS: Thank you for your observation. The authors have deleted TRIMP in this sentence.

REVIEWER: line 271: what is the CV capacity test – this is the first time this has been referred to as this.

AUTHORS: Thank you for your observation. The authors have changes CV test for aerobic power-capacity test.

REVIEWER: line 289: compared to EXP – please amend here and throughout.

AUTHORS: Thank you for your observation. The authors have included compared to HIGH and have reviewed this throughout the text.

REVIEWER: figure 3 and 4 – as of above major comments, I think the stats in which these are based are not well justified. With the restructure proposed, I will leave you to decide if these stay or go. If you have good justification for them/the analysis, please alter the key in these figures as I cannot tell which is LOW and which is EXP.

AUTHORS: Thank you for your recommendation. As in previous point we have said we believe that figures 3 and 4 are very important to show the % differences. We have done this analysis in a multitude of published manuscripts so that the reader can clearly see what the percentage change was during the study. Tables 2 and 3 show the absolute values in each group in T1 and T2. Do not hesitate, if the reviewer insists that we should delete that data, we would be happy to do so in the next round of reviews.

REVIEWER: line 388: I think this is a stretch to claim they were faster. I would focus only on the recovery/TRIMP which I believe is a nice finding – can you expanded on this specifically, and why this has occurred?

AUTHORS: Thank you for your interest. Due to the restructuring carried out in the discussion, this comment is included in it.

Round 2

Reviewer 1 Report

The authors have addressed all my comments.

Reviewer 2 Report

Thank you to the authors for responding to prior comments and adopting suggestions where necessary. I have some comments and suggestions to refine the paper. My main comments are still around the discussion and presentation of data that could be improved. 

First, the authors have adopted statistical advice from reviewer 1 to utilise non-parametric tests. However, now the authors no longer examine any group x time interaction effect. The analyses now just compare group effects for pre (randomised groups, therefore should be the similar), post, and then the change data (for variables this is calculated), and then compares within-group effects of time (for pre-post variables) by WSR test. The authors also chose not to add a within-group cohen’s d effect size analysis. This suggestion was purely to enhance this paper and provide a statistic to understand the magnitude of changes (pre to post), rather than just further probability testing. Although the current approach the authors have is not the approach I would use, I’ll leave it with the authors to decide what is correct and best for their work.

Second, given the authors have decided to keep the % change method of presenting and also analysing the data (Figs 2-4), can you please present the individual responses? This will be an easy change to the figures, but will provide some meaningful difference to the data that’s already presented in table form, and will likely provide a very good discussion point that I would like to see included as a new paragraph in the discussion. Were all responses for 120g/h group positive? This presentation will allow the reader to see, and your discussion will be insightful. Also, in regard to Figures 2-4, my understanding is that a Mann Whitney U (as the authors have stated they’ve used in the footnotes) is used to compare 2 groups only – therefore should this data be analysed by a Kruskal-Wallis test?

Third, regarding the results between lines 311-332, please reword some of these statements, as I’m sure you would agree there is not any “significant improvement in HIGH” (line 313, 315) in regard to % change responses after the marathon. Rather, as you state in line 317 for HST, there is a ‘smaller decline’ or you could state ‘trivial’ or no differences - the differences are with the LOW and MOD groups. The 0.5-1.2% ‘increase’ in performance for HIGH will be within the variability of this test I imagine.

Finally, the discussion is improved but there are still sections that do not provide insight to your data from this paper. For example, I am particularly interested in why you think TRIMP was lower in the high CHO group? You have shown here that athletes with higher CHO intakes have a lower impulse/stimulus on the system during the race (according to HR response). This is an interesting finding but not discussed (just mentioned in passing at the end of a paragraph). Was it the lower HRmean in the HIGH group? Was this because there was less CV drift in the HIGH group? Why might higher CHO intakes facilitate this response? This point, as well as the discussion on individual responses will be very interesting.

To make room for these discussion points, there are sections of the discussion that are directly replicated (or very close) from the introduction (lines 356+ & 378+ for example). It’s very noticeable when you read the paper front to back and these sections can be removed or altered. Again, as mentioned in first review, this second paragraph of your discussion also doesn’t mention your data until the last few sentences. Given this background detail is in the introduction, this whole paragraph could really be removed to make room for the above discussion of your data.  

Minor comments:

  • line 28: high intensity run capacity is here termed aerobic power capacity. Can you please keep consistent with either high intensity run capacity, or high aerobic power capacity, throughout the manuscript? Happy for you to choose which you prefer as long as it’s used consistently throughout! There are several instances where these are interchanged which is difficult for a reader.
  • line 32: instead of “internal load was significantly lower”, can you state TRIMP? This clears up method (HR-based) of internal load measurement for the reader.
  • line 34: “and in terms of fatigue”, can this be replaced with “run time” or “run capacity”?
  • line 53: this section is improved by your addition. However, internal load does not necessarily quantify fatigue? I would say that internal load is the ‘stimulus’ placed on the system, and if that’s reduced, resulting fatigue may be lower? Fatigue is assessed by your pre-post-tests. This is just a bit unclear.
  • line 68: over what time frame is this (1.23 g/kg) supplementation? Without a duration (I’m not sure of the 15 min frequency, is this maybe every 15 min across the exercise?) there’s no context to this statement.
  • line 83: ‘glycogen replenishment’ – suggest adding “and/or sparring”. The paper has a theme that the lower damage you have shown in your prior paper means glycogen is more quickly replenished after exercise, and that results in better recovery of NMF. However, given you don’t measure glycogen, could it be that you are sparing some (or at least keeping some for longer) given he high CHO intake? Maybe not given the duration of exercise, but we really don’t know.
  • line 98: this paragraph was justified by the authors as necessary to understand time course of recovery kinetics. I accept the rationale, but currently, this paragraph ends without really telling the reader why this paragraph is important. It needs a summary sentence that pulls it all together. For example: “Therefore higher CHO provision during exercise may spare glycogen… maintaining Ca release rates….. facilitating a quicker recovery ????”. This is for you to word, but without a summary to bring all these studies and data presented together, I still don’t think the paragraph adds much.
  • line 114: please add the word ‘reducing’ in the following sentence: …by ‘reducing’ internal exercise load…. and remove “long term” in this same sentence (and through the manuscript). Given in the time course paragraph you talk about studies showing up to 9 days recovery may be required, 24h is not long term?
  • line 154-155: could you remove the end of this sentence “with runners needing to consume ¼, 1/3 or ½ of the total g of CHO per hour according to their group (HIGH, MED, LOW, respectively)”. The method is clearer without this part of the sentence.
  • Figure 1 still states VO2max test – please amend.
  • line 193: suggest removing “in order to replenish glycogen levels to almost 90-93% of previous muscle glycogen [25,43].” As in your introduction, you discuss how EIMD can alter glycogen resynthesis rates. Given you don’t measure glycogen, this is speculative (particularly for a method section) within this timeframe.
  • line 198: Drop jump test is still stated here, but not mentioned elsewhere – please delete.
  • line 199: aerobic capacity test mentioned here – please keep consistent with high intensity run capacity test or similar throughout the entire manuscript.
  • line 216: now we have clarified there as a 1RM test completed, can you outline the protocol here please?
  • line 225: can you state reliability of measure from this APP here if you have the data?
  • line 241: this section where you have added part of the calculation used for TRIMP is improved and it’s now clear that you’ve used the original Banister method. However, can you please write the whole equation, in full, including the ‘adjustment’ factor. Is the equation you have used as cited? (Borresen and Lambert 2008). This is important information given the many ways this may be calculated. Can you please amend and display the equation in full as you have for MM on line 260? That would be helpful.
  • line 287-290: can you please remove data from text? The data is right below in the table and it’s easier to read/interpret without the actual data in line.
  • line 416: there is no correlation reported?
  • line 480: 3 participants withdrew due to GI disturbances – which groups were they in? This is important given the discussion here.